# Clinical and Radiographic Outcomes of Zirconia Dental Implants—A Clinical Case Series Study

**DOI:** 10.3390/ma15072437

**Published:** 2022-03-25

**Authors:** Jordi Gargallo-Albiol, Karl Böhm, Hom-Lay Wang

**Affiliations:** 1Oral and Maxillofacial Surgery Department, Universitat Internacional de Catalunya, 08017 Barcelona, Spain; 2Department of Periodontics and Oral Medicine, School of Dentistry, University of Michigan, Ann Arbor, MI 48109, USA; homlay@umich.edu; 3Clínica Dental, 08036 Barcelona, Spain; karl@dr-bohm.com

**Keywords:** zirconium, zirconia, ceramics, dental implants

## Abstract

The purpose of this clinical series was to evaluate the clinical and radiographic outcomes of 20 zirconia dental implants, with a minimum follow-up of two years. Patients with at least one zirconia dental implant were included, with a mean follow-up of 34.05 months. The patient complaints such as pain and foreign body sensation, as well as clinical parameters including peri-implant infections with suppuration, implant mobility, gingival index (GI), modified plaque index (mPI), modified sulcus bleeding index (mBI), probing depth (PD) and radiological distance between the implant shoulder to the closest mesial and distal bone-to-implant contact (BIC), were assessed. All zirconia implants were successfully integrated without any pain or foreign body sensation. No mobility was detected in any of the 20 implants. Clinical examination revealed a mean PD of 2.56 mm and a radiological mean distance between the implant shoulder and the initial site of visible bone-to-implant contact (BIC) of 1.44 mm. In addition, GI, mPI and mBI indicated minimal to no inflammation. Results obtained from this series suggest that one-piece zirconia dental implants achieve good clinical and radiographic outcomes over a mean follow-up of 34 months and may be deemed a good option in patients with favorable bone conditions.

## 1. Introduction

Since the 1970s, titanium (Ti) has been one of the most well-documented biomaterials in dental implantology. Due to its bioinert nature, strength, superiority over other metals in reducing Tresca stress and ability to osseointegrate, it has been regarded as the best material for manufacturing dental implants [1]. However, Ti does pose some challenges, such as a poor aesthetic appearance especially in areas of insufficient gingival and buccal bone thickness. Although rare, Ti has also been associated with allergic reactions [2]. Furthermore, Ti implants may generate some Ti particles which have been noted in peri-implant tissue. These Ti particles have recently been linked to the development of peri-implantitis [3] as well as to soft tissue lesions [4,5]. Moreover, Ti particles and ions could lead to the release of proinflammatory cytokines such as TNF-α and IL-1β, thereby playing a possible role in the inflammatory response [6,7,8]. Consequently, the need to establish an alternative material to Ti has been suggested.

Zirconium dioxide (ZrO_2_) or zirconia (Zr) was initially introduced as an alternative dental implant material due to the superior aesthetic effect of its color, particularly in areas characterized by a thin tissue phenotype [9,10], and it has also been related to a lesser tendency to accumulate plaque compared to Ti implants [10,11,12]. In addition, Zr implants show a 74–98% implant survival rate over 12 to 56 months and a 79.6–91.6% success rate over 6 to 12 months [13].

Surface characteristics are among the primary influencing factors of the osseointegration process [14,15], and there have been important developments in Ti implant surfaces over the years. Similarly, several preclinical investigations have demonstrated that Zr implants show similar bone-to-implant contact (BIC) and equivalent removal torque as Ti implants [16,17,18]. Zirconia surface modifications via acid etching and the creation of a microrough surface seem to enhance the biological properties of Zr implants, stimulating osseointegration and being comparable to the modified Ti surface [19,20]. Nonetheless, Zr dental implants seem to require longer healing times compared to titanium implants [21].

Furthermore, Zr dental implants can be found in one-piece or two-piece modalities. One-piece implants include the transepithelial abutment in the same implant fixation unit, avoiding the implant–abutment microgap and canceling transepithelial micromovements [22]. In contrast, one-piece implants exclude the possibility of abutment modification and make it necessary to perform cemented restorations.

According to Cionca et al. [23], some patients are aware of the existence of titanium dental implant alternatives on the market, and professionals must be prepared to respond to the patient demands. In a systematic review and meta-analysis, Elnayef et al. [24] concluded that in specific clinical scenarios, particularly in esthetic areas with a thin tissue phenotype, a Zr-based implant may be a good alternative to a Ti implant. However, Gahlert et al. [25] reported an unacceptable survival rate of 82.4% at three years with one-piece zirconia dental implants, and some studies involving one-piece Zr implants do not have follow-up periods of more than a year—though good results have been reported in terms of bleeding and plaque index, probing depth and marginal bone loss [26,27]. In contrast, few studies have evaluated Zr implants beyond one year of follow-up [28].

Thus, further studies are needed to ascertain Zr dental implant behavior involving rough surfaces and one- or two-piece modalities, with more than one year of follow-up.

The aim of the present case series was to document the clinical and radiographic outcomes of one-piece zirconia dental implants in healthy patients with favorable anatomical bone conditions over at least two years of follow-up after final restoration.

## 2. Materials and Methods

### 2.1. Sample Description

The study protocol was approved by the Clinical Research Ethics Committee of the Universitat Internacional de Catalunya (Study code: CIR-ELC-2021-10).

The inclusion criteria were one-piece Zr implants (Straumann^®^ Pure Ceramic implant system; Straumann Holding AG, Basel, Switzerland) placed with an either early or delayed approach [29] and with a minimum follow-up period of two years after final crown delivery. Healthy patients were included, with optimal bone conditions (≥6 mm bucco-lingual bone width and more than 8 mm vertical length) for adequate primary implant stability. Guided bone regeneration (GBR) was allowed to enhance peri-implant bone contour if needed and was always performed simultaneously using demineralized bovine bone grafts (Bio-Oss, Geistlich Pharma AG, Wolhusen, Switzerland) and a collagen membrane (Bio-Gide, Geistlich Pharma AG, Wolhusen, Switzerland). We excluded two-piece Zr implants, as well as implants other than Zr implants (including titanium implants), bone deficiencies requiring bone augmentation procedures before placing of the implants, cases of extensive bone regeneration procedures carried out simultaneously to implant placement and one-piece Zr implants definitely restored less than two years ago.

Dental records from January 2013 to December 2019 corresponding to patients who underwent one-piece zirconia implantation were carefully screened. Then, the eligible patient records were reviewed to assess follow-up status. Patients who fulfilled the inclusion criteria were invited by phone to participate in the study.

All subjects were aware that a ceramic implant was being placed and that regular check-ups would be made to assess the clinical and radiographic conditions of the implant. All of the patients provided informed consent before participating in the study, and the ALARA (as low as is reasonably achievable) radiographic principles were followed.

### 2.2. Surgical Procedures

Digital 3D cone-beam computed tomography (CBCT) presurgical planning (Orthophos XG, Dentsply Sirona, York, PA, USA) was made for each patient. All patients were treated using a full thickness envelope flap with a crestal incision in the center of the alveolar ridge and with a marginal incision around the adjacent neighboring teeth in order to sufficiently expose the crestal bone of the edentulous space. Following this, a conventional drilling sequence was performed according to the instructions of the manufacturer (Straumann^®^ Pure Ceramic implant system, Straumann Holding AG, Basel, Switzerland). The bone drilling sequence involved an initial stainless steel round drill 2.3 mm in diameter, followed by twist drills measuring 2.2 and 2.8 mm in diameter. In the case that a 4.1 mm diameter implant had to be placed, an additional 3.5 mm diameter twist drill was also used. An indicator of the direction of the bone ostectomy (and height of the pillar) was used after the 2.8 mm and 3.5 mm diameter drills to ensure proper positioning of the implant and adequate height of the abutment. Finally, the bone drilling sequence was completed with a stainless-steel profile drill for the 3.5 or 4.1 mm diameter implants. A ceramic implant measuring 8 or 10 mm in length and 3.3 or 4.1 mm in diameter was then installed. Figure 1 illustrates the entire surgical and prosthodontic workflow, and a detailed intraoperative view of a Zr monotype implant placement before flap closure is shown in Figure 2.

Afterwards, GBR procedures were performed if needed, a plastic abutment protector was placed, and flaps were closed with single sutures to ensure transmucosal implant healing. An intraoral radiograph was taken immediately after surgery (Figure 3). Postoperative medication was then prescribed in the form of amoxicillin 500 mg (or clindamycin 300 mg in patients allergic to penicillin) every 8 h for 7 days, ibuprofen 600 mg every 8 h for four days, and chlorhexidine rinses twice a day for 15 days.

### 2.3. Prosthetic Restoration

Immediately after implant placement surgery, conventional impressions using the manufacture impression copings were performed. A laboratory-processed resin was used to fabricate a temporary crown and was inserted to protect the implant abutment and modulate the gingival contour (Figure 4).

The temporary crown was placed in infraocclusion, ascertaining the absence of occlusal contacts in maximum intercuspidation and during lateral mandibular movements. The gingival implant contours were further adapted by chair-side modifying of the temporary crown. Three months after implant placement, definitive impressions were made using conventional manufacture impression copings. An acrylic try-in was tested and, after laboratory finishing and personalizing, a monolithic zirconia crown was finally cemented using an elastomeric resin cement (Premier^®^ Implant Cement™, Plymouth, MA, USA). Excess cement was carefully removed. Annual follow-up appointments were then scheduled to perform the clinical and radiographic measurements. Figure 5 and Figure 6 illustrate a definitive ceramic nonmetal restoration and the corresponding intraoral X-ray image, respectively, after three years and three months of follow-up.

Lastly, Figure 7 and Figure 8 show a definitive ceramic restoration in another case located in the maxillary left first premolar space and its corresponding intraoral X-ray view, after three years of follow-up.

### 2.4. Clinical and Radiographic Parameters

Clinical parameters were considered based on the Buser criteria [1]: (a) presence of subjective complaints related to the area of treatment; (b) recurrent infections in the peri-implant region, accompanied by suppuration; (c) mobility of the implant fixture (0 or 1 corresponding to no or yes, respectively); (d) gingival index (GI) at four sites around the implant (mesial, distal, buccal and lingual); (e) amount of plaque detected on the implant via the modified plaque index (mPI), at four sites around the implant (mesial, distal, buccal and lingual); (f) amount of sulcus bleeding according to the modified sulcus bleeding index (mBI), taken at two sites around the implant (buccal and lingual) and (g) peri-implant probing depth (PD) at four sites (mesial, distal, buccal and lingual), measured in millimeters using the North Carolina periodontal probe (Hu-Friedy Mfg. Co., Chicago, IL, USA). The peri-implant indexes (GI, mPI and MBI) are detailed in Table 1.

Additionally, radiological examination comprised an intraoral radiograph of the region of interest using the paralleling technique and the Rinn-type film holder (Kerr Corporation, Bioggio, Switzerland). To evaluate the distance between the implant shoulder and the closest mesial and distal BIC, a calibrated image using special software (Sidexis, Dentsply Sirona, York, PA, USA) was employed. Intraexaminer calibration was performed by taking all measurements two times on different days [30]: the mean of the mesial and distal distance values was calculated per implant.

Thus, each included implant was categorized as success or failure using the collected clinical and radiographic data, based on the predetermined success criteria suggested by Buser et al. [1].

## 3. Results

### 3.1. Sample Analysis

A total of 20 dental implants (seven in women and 13 in men) in 10 healthy patients were included in the study. The mean patient age was 60.75 years (range 45–79). Eight patients were nonsmokers, while two patients smoked <5 cigarettes per day. Five implants (25%) were 3.3 mm in diameter and 15 (75%) were 4.1 mm in diameter. Eleven implants were placed in the maxilla (55%) and nine in the mandible (45%). All implants were located in the posterior area (six in premolars and 14 in molars): five in superior premolars (two in the right second premolar and three in the left first premolar); six in the superior molars (two in the right first molar, one in the right second molar, two in the left first molar and one in the left second molar); one in the second left premolar and eight in the inferior molars (three in the right first molar, one in the right second molar, three in the first left molar and one in the second left molar). All implants achieved primary stability and were installed with an insertion torque of at least of 30 Ncm. Additionally, guided bone regeneration was employed simultaneously to implant placement in 12 implants (60%) in order to enhance the implant bone contour. The mean follow-up was 34.05 months (range 24–84). Overall, the definite prosthetic restoration was placed after an average of 2.67 months (range 1.5–7). The detailed sample data are provided in Table 2.

### 3.2. Clinical and Radiological Outcomes

All implants were classified as successful according to the criteria of Buser et al. [1], with no patient-reported complaints of pain, foreign body sensation or dysesthesia. Implant mobility was not observed in any implant. The mean gingival index, modified plaque index and modified sulcus bleeding index values were 0.38, 0.33 and 1.075, respectively, indicating scarce gingival inflammation and plaque accumulation. A peri-implant soft-tissue bleeding score of close to 1 also represents little bleeding. Nonetheless, bleeding signs were observed in some cases (Table 3). The overall mean probing depth (PD) was 2.56 mm (range 0–5). Lastly, the overall radiological mean distance between the implant shoulder and the initial site of visible bone-to-implant contact (BIC) was 1.44 mm (range 0.17–2.94). The detailed results are reported in Table 3.

## 4. Discussion

Zirconia implants have been shown to be a plausible alternative to Ti implants due to their biocompatibility and resistance to fracture and compression, being lighter in color and featuring considerably lower bacterial adherence [10,31]. However, some authors have reported lower implant survival rates with Zr implants in comparison to Ti implants [32]. Manzano et al. (2014) showed that a treated Zr implant surface could increase the BIC values, resulting in a reduced occurrence of early failure and reversal torque commensurate versus that of Ti implants [12]. In addition, a meta-analysis [24] has reported Zr implants to have a 91.5% survival rate and a 91.6% success rate over a mean follow-up period of 42.37 months. This study also suggested that Zr-based dental implants may be advantageous in thin gingival phenotypes, especially in the aesthetic zone, compared to Ti implants, due to their white color [33]. The premolar area is less challenging than the canine or incisor zone for one-piece implants, due to the lesser aesthetic demands involved. Two-piece implants may result in a better aesthetic outcome.

Our clinical series displayed optimal clinical and radiological results. All the implants exhibited adequate bone levels and healthy peri-implant tissues at final assessment. These findings are in line with the data presented by Borgonovo et al. [26] and Vilor-Fernández et al. [27], who reported no bleeding, minimal plaque index and hard and soft tissue stability after at least 6 months or 1 year of follow-up, respectively, with one-piece Zr implants, and the clinical attachment level of implants of this kind was seen to remain stable after three years of follow-up, as reported by Balmer et al. [28].

Thus, evaluation of the peri-implant soft-tissue parameters was certainly positive, supporting the observations of previous authors regarding effective adaptation at the gingiva–Zr interface [34]. Furthermore, the absent or scarce plaque accumulation and little sulcus bleeding observed across the cases may be explained by the previously demonstrated lesser bacterial attachment related to Zr implants [35]—ultimately resulting in reduced inflammation and a lesser prevalence of mucositis and/or peri-implantitis as compared to traditional Ti implants. However, these results are to be interpreted with caution, due to the limited sample size and considering that all implants were being evaluated in healthy patients with good oral hygiene and suitable bone volume. In addition, in order to avoid deficient analysis as a consequence of the small data differences found in the clinical and radiological parameters, comparisons of baseline and follow-up measurements were not contemplated.

The overall mean PD of 2.56 mm found in this series may certainly be attributed to the distinct microdesign of the implant, involving a rough surface with a 1.8 mm smooth coronal surface. This specific tissue-level implant design has formerly displayed positive tissue stability in long-term studies [36]. These PD outcomes are slightly lower than those reported by Borgonovo et al. [26] and Balmer et al. [28], who found mean PDs of 3.19 mm and 3.52 mm after at least 6 months or 3 years of follow-up, respectively, both using the same one-piece Zr implant. Furthermore, we must take into account that PD of one-piece Zr implants tends to increase over time, according to different studies [27,28].

All these findings are consistent with the radiographic parameters. The radiological mean distance between the implant shoulder and the initial site of visible bone-to-implant contact was 1.4 mm, indicating good stability of the marginal bone. One-piece implants have been reported to exhibit marginal bone stability superior to that of a two-piece implant by avoiding the implant–abutment microgap and micromovements [22]. Rodriguez et al. [37] combined one- and two-piece Zr implants and recorded similar clinical and radiological outcomes, though with a lower success rate.

With regard to marginal bone loss (MBL), the present study does not address this issue, but Borgonomo et al. observed a MBL of 1.2 mm after four years of follow-up with one-piece Zr implants [38], and Balmer et al. [28] reported 0.72 mm of MBL from implant insertion for three years of follow-up after the final restoration. Additionally, Elnayef et al. reported 0.14 mm more MBL with Zi versus Ti implants [24].

The limitations of our study include (but are not limited to) its small sample size and the lack of comparison with the baseline parameters. Patient recruitment was limited to the cases that fulfilled the inclusion criteria, and this certainly may have had an impact upon data compilation. Although such recruitment implied good homogeneity regarding the general patient conditions and local bone environment, it would be strongly advisable to have a larger sample size. Additionally, some cases were treated with GBR while others were not; one-piece implants rendered abutment modification impossible, this being a major challenge in esthetic zones; and lastly, the cemented restorations with cement margins were determined by the manufacturer. On the other hand, low temperature degradation of zirconia is also a debated phenomenon, though some investigations indicate that ceramics and stabilized zirconia are favorable materials for use in the oral environment in this regard [39,40]. Thus, the results presented in our case series should be interpreted with caution, and larger randomized controlled clinical trials are needed to evaluate the long-term clinical suitability of Zr implants.

## 5. Conclusions

One-piece zirconia implants seem to be a feasible option in patients with favorable bone anatomy, offering optimal clinical and radiographic outcomes after a mean follow-up period of 34.05 months (range 24–84), showing hard and soft tissue stability at the last follow-up visit. Nevertheless, further research is imperative in order to evaluate the long-term performance of zirconia implants, involving challenging situations such as bone deficiencies and considering the peri-implant clinical and radiologic parameters and the biomechanical behavior of both two-and one-piece zirconia implants.

## Figures and Tables

**Figure 1 materials-15-02437-f001:**
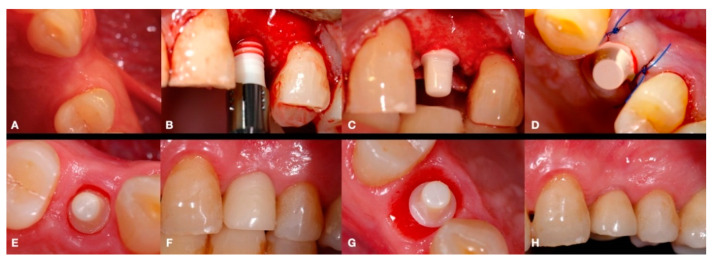
Surgical and prosthodontic workflow: (**A**) preoperative occlusal view; (**B**) one-piece Zr implant placement; (**C**) Zr transepithelial abutment image before flap closure; (**D**) immediate postoperative view of the abutment plastic protector and suture; (**E**) occlusal view of the Zr transepithelial abutment before temporary restoration; (**F**) lateral view of the temporary restoration; (**G**) occlusal view of the Zr transepithelial abutment before definitive restoration (note the soft tissue contour modulation using the temporary crown); (**H**) lateral view of the final restoration.

**Figure 2 materials-15-02437-f002:**
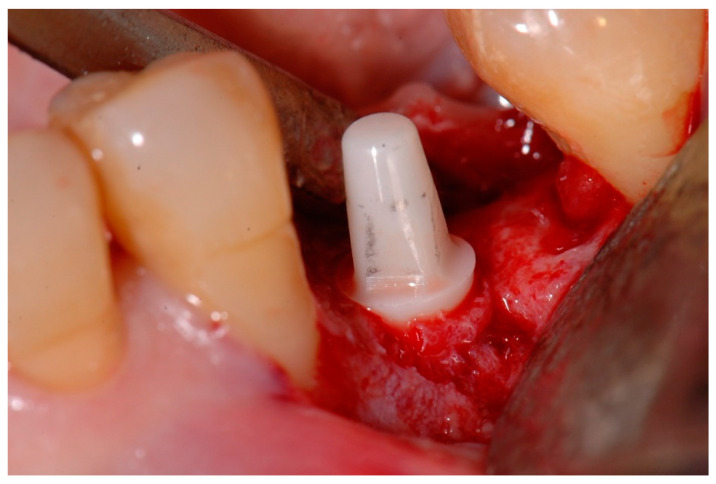
Intraoperative view of a one-piece zirconia implant placed in the mandibular left second premolar position.

**Figure 3 materials-15-02437-f003:**
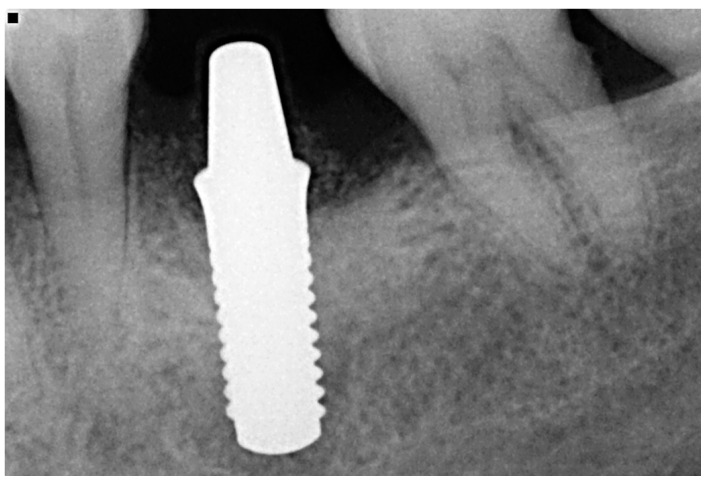
Immediate postoperative intraoral X-ray view after one-piece zirconia implant placement.

**Figure 4 materials-15-02437-f004:**
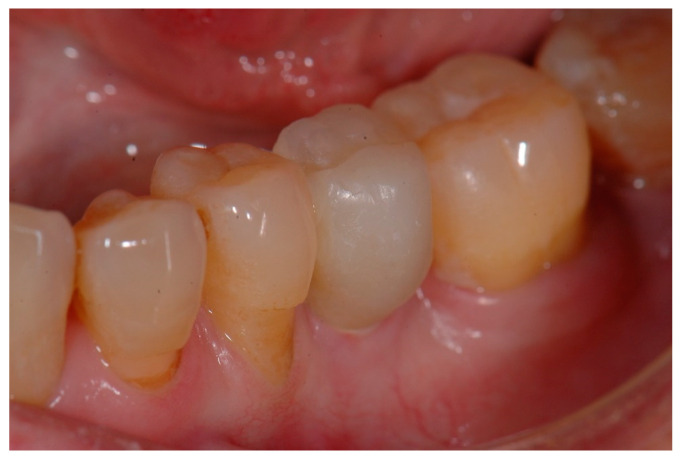
Laboratory-made long-term temporary crown performed to modulate the gingival contour.

**Figure 5 materials-15-02437-f005:**
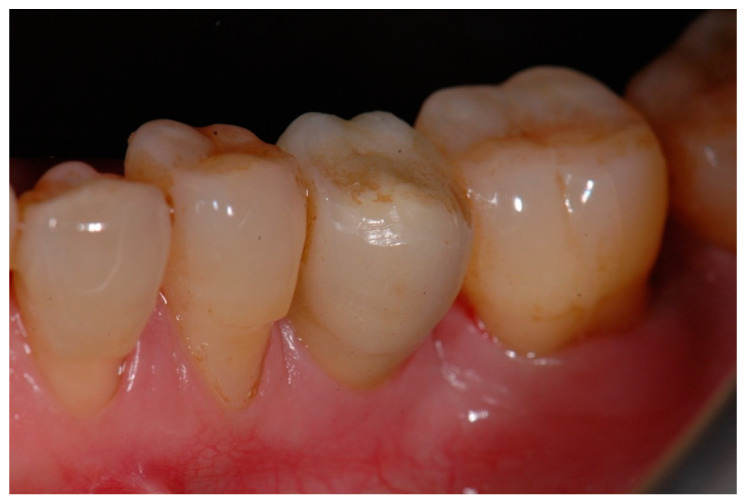
Definitive ceramic restoration (at 3 years and 3 months of follow-up).

**Figure 6 materials-15-02437-f006:**
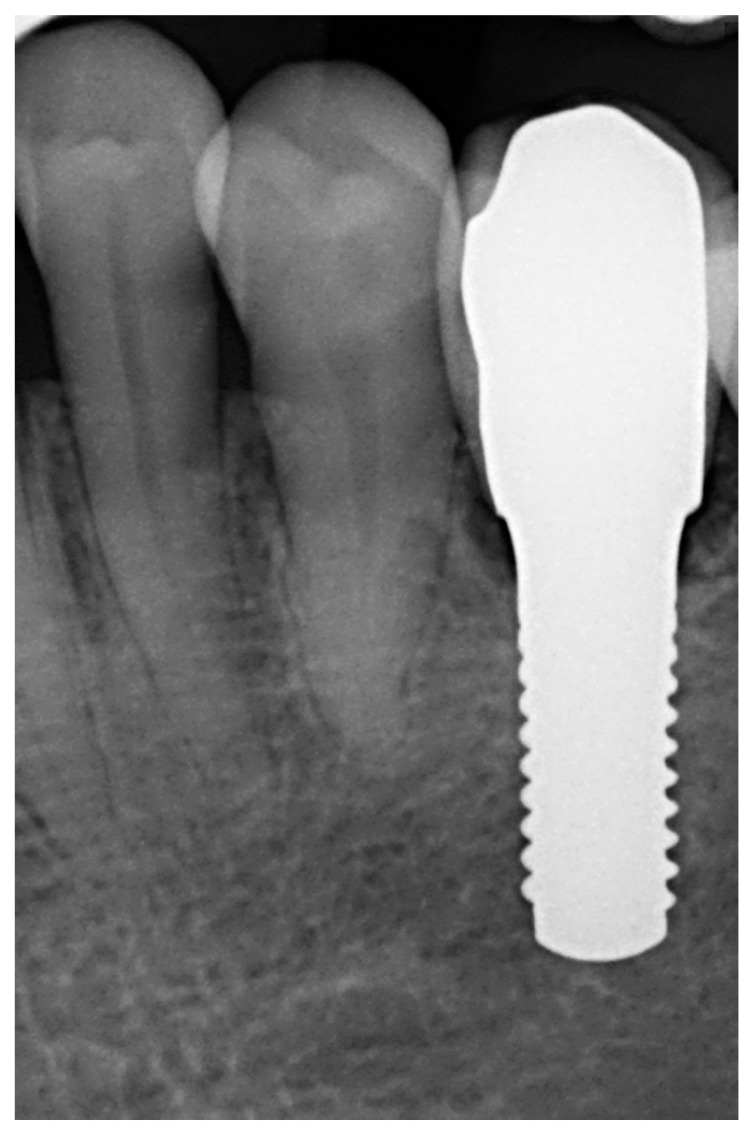
Intra-oral X-ray image after 3 years and 3 months follow-up.

**Figure 7 materials-15-02437-f007:**
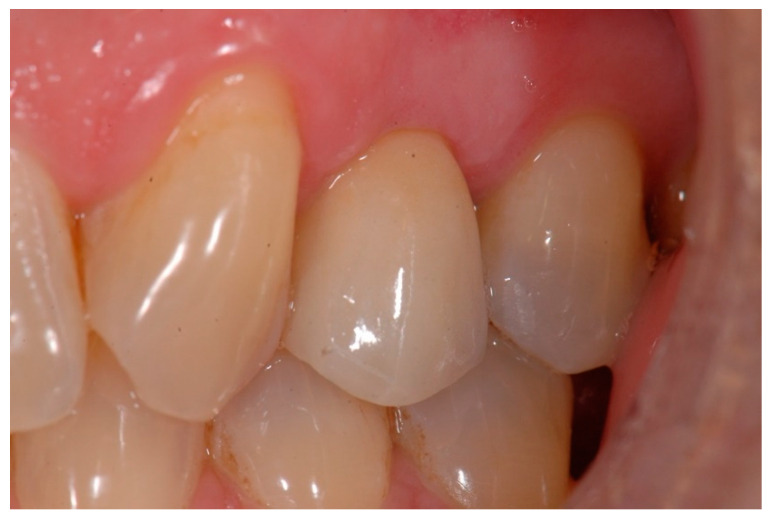
Definitive ceramic restoration in a maxillary left first premolar (after 3 years of follow-up).

**Figure 8 materials-15-02437-f008:**
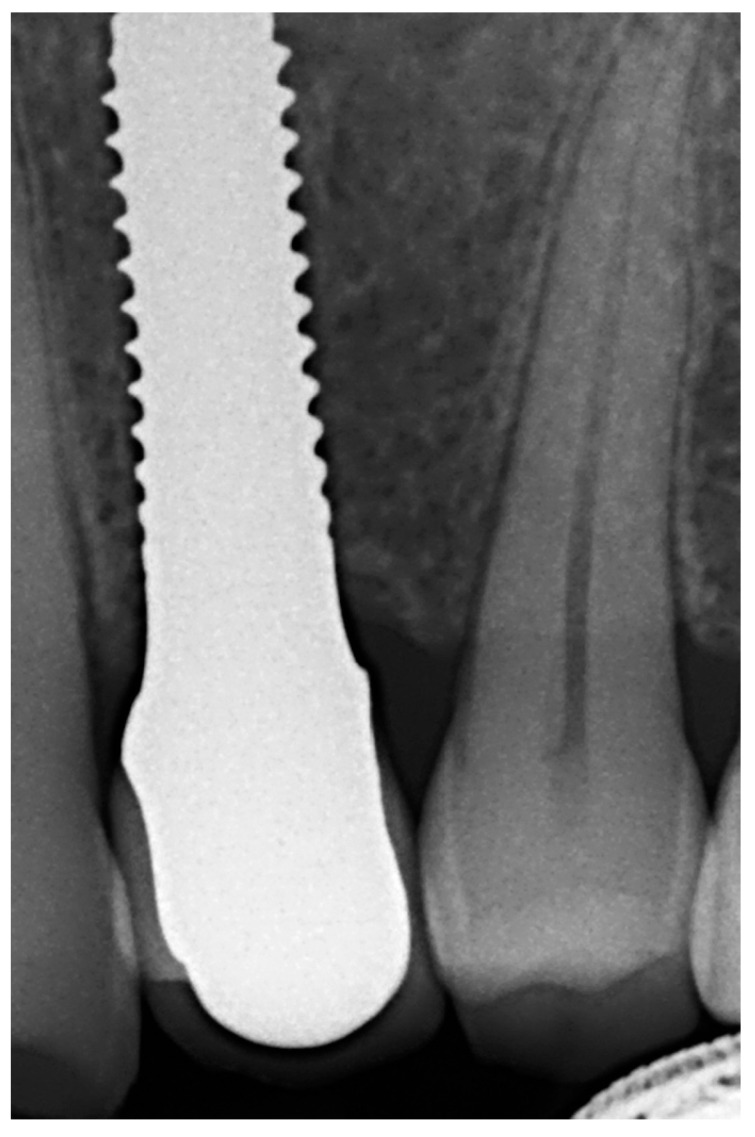
Intraoral X-ray view after three years of follow-up.

**Table 1 materials-15-02437-t001:** Peri-implant index details based on the Buser criteria [1].

N	Index Score	Parameters
Gingival index (GI)	1	Normal gingiva: natural coral pink with no inflammation
	2	Mild inflammation: slight changes in color, slight edema. No bleeding on probing
	3	Moderate inflammation: redness, edema and glazing
Modified plaque index (mPI)	0	No detection of plaque
	1	Plaque only recognized by running a probe across the smooth marginal surface of the implant
	2	Plaque can be seen by the naked eye
	3	Abundance of soft matter
Modified sulcus bleeding index (mBI)	0	No bleeding
	1	Isolated bleeding spots visible
	2	Blood forms a confluent red line on margin
	3	Heavy or profuse bleeding

**Table 2 materials-15-02437-t002:** General overview and characteristics of the included cases.

N	Gender	Age	Smoking (>10 cig/Day)	Implant Position	Implant ∅ (mm)	Implant Length (mm)	Bone Augmentation	Definitive Restoration Placement (Months)
1	M	70	Y	15	4.1	10	N	2.5
2	M	70	Y	26	4.1	10	Y	2.5
3	F	47	N	36	4.1	12	Y	3
4	F	47	N	46	4.1	12	Y	3
5	M	50	N	16	4.1	8	N	3
6	M	50	N	46	4.1	10	N	3
7	F	45	N	36	4.1	10	N	2.5
8	M	63	N	17	4.1	10	N	1.5
9	M	63	N	16	3.3	10	Y	1.5
10	M	63	N	27	4.1	10	N	1.5
11	M	63	N	26	4.1	10	Y	1.5
12	M	63	N	37	3.3	10	Y	1.5
13	M	63	N	36	3.3	10	Y	1.5
14	M	63	N	47	3.3	10	Y	1.5
15	M	63	N	46	3.3	10	Y	1.5
16	F	61	N	24	4.1	12	N	3
17	M	75	N	24	4.1	12	N	4
18	F	48	N	15	4.1	12	Y	4
19	F	69	N	35	4.1	10	Y	4.0
20	F	79	N	24	4.1	10	Y	7.0
Mean		60.75						2.67

**Table 3 materials-15-02437-t003:** Clinical and radiological outcomes (GI: gingival index; mPI: modified plaque index; mSBI: modified sulcus bleeding index; PD: probing depth; B: buccal; L: lingual; M: mesial; D: distal; BIC: bone-to-implant contact).

N	Follow-Up (Months)	GI (B, L, M, D)	Mean GI	mPI (B, L, M, D)	Mean mPI	mSB (B, L)	Mean mSB	PD (B, L, M, D)	Mean PD	Implant Shoulder-First BIC (M; D)	Mean Implant Shoulder-First BIC
1	24	0, 0, 1, 0	0.25	0, 0, 0, 0	0	0, 0	0	0, 0, 0, 2	0.5	0.54; 0.17	0.355
2	24	0, 1, 0, 1	0.5	0, 0, 0, 0	0	0, 1	0.5	0, 2, 3, 4	2.25	1.8; 2.1	1.95
3	24	0, 2, 0, 1	0.75	0, 1, 2, 2	1.25	2, 1	1.5	3, 4, 3, 4	3.5	2.2; 2.2	2.2
4	24	0, 0, 2, 1	0.75	0, 0, 1, 1	0.5	0, 1	0.5	2, 3, 5, 4	3.5	2.3; 2.4	2.35
5	24	0, 0, 1, 0	0.25	0, 0, 1, 1	0.5	0, 1	0.5	1, 2, 1, 1	1.25	1.53; 1.84	1.685
6	24	0, 0, 0, 1	0.25	0, 0, 1, 1	0.5	0, 1	0.5	1, 2, 1, 2	1.5	2.32; 2.15	2.235
7	24	1, 0, 1, 0	0.5	1, 1, 0, 0	0.5	0, 0	0	2, 1, 4, 3	2.5	1.66; 2.01	1.835
8	24	0, 0, 1, 0	0.25	1, 0, 1, 0	0.5	2, 3	2.5	4, 2, 3, 2	2.75	0.81; 0.9	0.855
9	24	0, 1, 0, 0	0.25	0, 1, 0, 1	0.5	2, 2	2	3, 3, 2, 2	2.5	0.45; 0.29	0.37
10	24	0, 0, 0, 1	0.25	0, 0, 0, 0	0	2, 1	1.5	4, 3, 5, 3	3.75	2.2; 1.49	1.845
11	24	0, 1, 0, 0	0.25	0, 1, 0, 0	0.25	2, 2	2	4, 4, 3, 2	3.25	0.15; 2.88	1.515
12	24	0, 0, 1, 1	0.5	0, 1, 1, 1	0.75	1, 3	2	4, 3, 4, 4	3.75	0.43; 0.35	0.39
13	24	1, 0, 1, 1	0.75	0, 0, 0, 1	0.25	2, 3	2.5	4, 4, 4, 4	4	2.75; 0.92	1.835
14	24	0, 0, 0, 2	0.5	0, 0, 0, 1	0.25	2, 3	2.5	5, 3, 3, 3	3.5	0.48; 0.74	0.61
15	24	0, 0, 1, 1	0.5	1, 0, 0, 1	0.5	2, 3	2.5	5, 4, 4, 3	4	2.2; 2.94	2.57
16	36	0, 1, 1, 1	0.75	0, 0, 0, 0	0	0, 0	0	2, 2, 3, 3	2.5	1.9; 2.1	2
17	72	0, 0, 0, 0	0	0, 0, 0, 0	0	0, 0	0	1, 2, 1, 1	1.25	1.8; 2.1	1.95
18	72	1, 1, 0, 0	0.5	0, 0, 0, 0	0	0, 0	0	1, 2, 1, 1	1.25	0.9; 1.8	1.35
19	84	0, 0, 0, 0	0	1, 0, 0, 1	0.5	1, 0	0.5	2, 1, 1, 1	1.25	1.5–1.8	1.65
20	57	0, 0, 0, 0	0	0, 0, 0, 0	0	0, 0	0	2, 3, 2, 3	2.5	0.3–0.4	0.35
Mean	34.05		0.38		0.33		1.075		2.56		1.44

## Data Availability

The data presented in this study are available upon request to the corresponding author.

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
