# Peer review of "Clinical and Radiographic Outcomes of Zirconia Dental Implants—A Clinical Case Series Study"

_materials, 2022, doi:10.3390/ma15072437_

Round 1
Reviewer 1 Report
The manuscript has the same text and structure as the second version revised on July 2021, except for a sentence relative to ethical approval. I confirm that the manuscript is suitable for publication.
Reviewer 2 Report
Dear Gargallo-Albiol et al.,
The manuscript “Clinical and radiographic outcomes of zirconia dental implants.” (materials-1614228) by Gargallo-Albiol et al. evaluate the clinical and radiographic outcomes of 20 zirconia dental implants, with a minimum follow-up of two years. The topic is interesting, but I think this article should reconsider after proper changes in major revision for publication in Materials. Some of my specific comments are:
- I would encourage and advise the authors to adopt some of the additional references published by MDPI in the introduction section as follow:
Tresca Stress Simulation of Metal-on-Metal Total Hip Arthroplasty during Normal Walking Activity. Materials (Basel). 2021, 14, 7554. https://doi.org/10.3390/ma14247554
- I see some errors on English in some areas of the present manuscript. To improve the quality of English used in this manuscript and make sure English language, grammar, punctuation, spelling, and overall style are correct, further proofreading is needed. As an alternative, the authors can use the MDPI English proofreading service for this issue. For example in subsection 2.2 Surgical procedure should be changed from “surgical procedure” to “Surgical Procedures”. Uppercase in Procedures is related in my comments number 4.
- It seems that in the present manuscript, the authors do not follow Materials, MDPI manuscript template properly and have a misunderstanding using the manuscript template. The authors can download published manuscripts by Materials, MDPI, and compare them with the present author's manuscript to ensure typesetting is appropriate.
- The author seems to have made an error in using uppercase and lowercase in the title of the present article and all of the subsections that should be corrected.
- The state of the art, the significance of the present study, and research novelty are not clearly present, the authors should highlight it more advanced in the introduction section.
- In the introduction section, the authors should explain the previous research conducted and its shortcomings. It will uphold the research gap that you filled with your research novelty. I recommend the authors elaborate their introduction section.
- In the materials and methods section, the authors should add one systematic figure to illustrate the workflow of experimental testing in the present study to make the reader more interested and easier to understand rather than only using dominant text and specific figures to explain.
- In subsection 2.2. Surgical procedure, the authors should explain the patient main specification involved in this research to give clearly understanding to the readers regarding correlation between the patients involved and the results obtained.
- The author must provide a detailed specification and use condition more detail regarding all tools used in the research carried out so that the reader can estimate the accuracy and differences in the results that the authors describe due to the use of different tools in future studies.
- In the Subsection 3.1. Sample analysis, the authors states “A total of 20 dental implants (7 in women and 13 in men) in 10 healthy patients were included in the study”. However, the authors do not explain it in the materials and methods section regarding sample used with 20 dental implants. The authors must state and explain regarding sample of 20 dental implants in the materials and methods section.
- What is the author's basis for using a sample of 20 dental implants? Are there procedures, literatures, or previous research that supports this? It should state, explain, and elaborate properly in the materials and methods section, bit if is it done following available samples that allow it to be tested in the current study, it should state and explain the limitation of the study in the end of the results section.
- In the results section, authors are advised to compare the results they obtain with previous similar/identical studies if it is possible.
- The conclusion of the present manuscript is not solid. Further elaboration is needed.
- Further research needs to be explained in the conclusion section.
I am pleased to have been able to review the author's present manuscript. Hopefully, the author can revise the current manuscript as well as possible so that it becomes even better. Good luck for the author's work and effort.
Best regards,
The Reviewer
Reviewer 3 Report
Thank you for the opportunity to review this interesting clinical study. This case series aimed to document one-piece zirconia dental implants' clinical and radiographic outcomes in healthy patients with favourable anatomical bone conditions over at least two years of follow-up.
The study is well done, but it would be good to emphasize in the title that it is about a clinical case series study.
Also, it is not a well-written part about the study population. Please authors to refine that part of the study. Was this survey reported on clinical trials? In what period were dental implants placed? In what positions were they placed? Were there inclusive or exclusive criteria?
Round 2
Reviewer 2 Report
Dear Gargallo-Albiol et al.,
After carefully reading the author's revised manuscript entitled "Clinical and radiographic outcomes of zirconia dental implants." (materials-1614228) by Gargallo-Albiol et al., The authors have been made significant improvements in the revised manuscript. Also, all of the issues in my review report have been addressed precisely.
With my pleasure, I recommend the manuscript should be accepted for publication on Materials.
Best regards,
The Reviewer